# Experimental Demonstration of an Efficient Mach–Zehnder Modulator Bias Control for Quantum Key Distribution Systems

**Chang-Hoon Park** [1,2]🆔**, Min-Ki Woo** [2]**, Byung-Kwon Park** [1]🆔**, Seung-Woo Jeon** [1]**, Hojoong Jung** [1]**, Sangin Kim** [2]🆔 **and Sang-Wook Han** [1,3,]*

1. Center for Quantum Information, Korea Institute of Science and Technology (KIST), Seoul 02792, Korea; originalpch@kist.re.kr (C.-H.P.); pbk2324@naver.com (B.-K.P.); sw_jeon@kist.re.kr (S.-W.J.); hojoong.jung@kist.re.kr (H.J.)
2. Department of Electrical and Computer Engineering, Ajou University, Suwon 16499, Korea; namdo6sung@hanmail.net (M.-K.W.); sangin@ajou.ac.kr (S.K.)
3. Division of Nano and Information Technology, KIST School, Korea University of Science and Technology, Seoul 02792, Korea
* Correspondence: swhan@kist.re.kr; Tel.: +82-031-546-7474

**Abstract:** A Mach–Zehnder modulator (MZM) is necessary for implementing a decoy-state protocol in a practical quantum key distribution (QKD) system. However, an MZM bias control method optimized for QKD systems has been missing to date. In this study, we propose an MZM bias control method using $N$ ($\geq 2$) diagnostic pulses. The proposed method can be efficiently applied to a QKD system without any additional hardware such as light sources or detectors. Furthermore, it does not reduce the key rate significantly because it uses time slots allocated to existing decoy pulses. We conducted an experimental demonstration of the proposed method in a field-deployed $1 \times 3$ QKD network and a laboratory test. It is shown that our method can maintain the MZM extinction ratio stably over 20 dB (bit error rate $\leq 1\%$), even in an actual network environment for a significant period. Consequently, we achieved successful QKD performances.

**Keywords:** quantum key distribution; Mach–Zehnder modulator; bias control; decoy-state protocol

## 1. Introduction

The rapid development of quantum computing technology [1–6] has radically stimulated interest in quantum cryptography. Particularly, quantum key distribution (QKD) systems, which allow two distant parties to share secure keys [7–13], have attracted considerable attention as a core element of quantum cryptography.

Practical QKD systems using weak coherent pulses are sensitive to environmental noise. Accordingly, many efforts [14–25] have been made to design a noise-tolerant optical architecture and to stabilize optical devices, such as lasers, detectors and modulators. However, to date, a Mach–Zehnder modulator (MZM) bias control method for QKD systems is missing, despite the MZM being essential for implementing a decoy-state protocol [26,27], which is the only way to prevent photon number splitting attacks [28,29], except for ideal single-photon sources. In other fields, there are primarily two types of MZM bias control methods: one that utilizes optical power monitoring [30–34] and the other that utilizes a dither signal [35–43]. In the former case, the input and output power or their ratio are used as the feedback signal. In the latter case, a dithering signal is used to generate the first- and second-order harmonics, and, subsequently, the bias voltage is controlled according to their power ratio. However, they are not suitable methods for a QKD system because even a small amount of optical crosstalk noise from the strong light used in their bias control may increase errors in the QKD system. For the same reason, commercial products for bias control requiring additional optical devices are not efficient

for QKD systems. In addition, they are incompatible with system miniaturization owing to additional optical devices.

In this study, we propose an MZM bias control method that can be efficiently applied to QKD systems. The proposed method does not require additional devices, such as lasers, beam splitters (BSs) or detectors. This is because it is implemented using a software modification that only adds $N$ ($\geq 2$) diagnostic pulses [34] for bias drift estimation. Therefore, it does not conflict with system miniaturization. Moreover, it does not degrade the key rate significantly because the number of signal pulses can be maintained regardless of the number of diagnostic pulses. This is performed by consuming the time slots allocated to existing decoy pulses. Such advantages are significant when implementing a QKD network system [44–49] comprising many users that require individual bias controllers. In addition, it can be efficiently implemented on a parallel processor such as a field-programmable gate array (FPGA) device. This is because its calculations can be parallelized. As the proposed method immediately compensates for the phase drift estimated by the diagnostic pulses, it has a higher convergence rate than conventional proportional–integral–derivative (PID) control, in which the current point converges to the desired setpoint by gradually reducing errors. Finally, we experimentally demonstrated the proposed method in a laboratory and in the field to show its feasibility. In the field test, we applied the control method to a testbed consisting of a $1 \times 3$ QKD network system installed in the security facility of a smart factory in South Korea. We verified that the method could keep the MZM bias point at the desired setpoint (null point) by compensating for the bias drift owing to environmental changes.

The remainder of this study is organized as follows. In Section 2, the proposed control method is described. In Section 3, we present the results of the laboratory and field tests. Finally, our main conclusions and findings are summarized in Section 4. See Table 1 for the abbreviations and symbols used in this work.

**Table 1.** Nomenclature of the symbols and abbreviations used in this article.

| Abbreviation | Description | Symbol | Description |
|---|---|---|---|
| BS | Beam splitter | $\theta_{drift}$ | Practical phase drift |
| CIR | Circulator | $\theta_{drift}^T$ | Theoretical phase drift, $\left[0^\circ, 360^\circ\right)$ |
| CW | Continuous-wave | $p_i$ | Practical detection probability |
| DC | Direct-current | $p_{T_i}\left(\theta_{drift}^T\right)$ | Theoretical detection probability |
| DFB | Distributed feedback | $\mathrm{Err}\left(\theta_{drift}^T\right)$ | Error between $p_i$ and $p_{T_i}\left(\theta_{drift}^T\right)$ |
| DL | Delay line | $\theta_{mod}$ | Phase modulation |
| DWDM | Dense wavelength division multiplexer | $\theta_{mod}^i$ | Phase modulation of the *i*-th diagnostic pulse |
| ER | Extinction ratio | $I_{out}$ | Output intensity |
| FM | Faraday rotator mirror | $L_{in}$ | Insertion loss |
| FPGA | Field-programmable gate array | $I_{in}$ | Input intensity |
| MCU | Microcontroller unit | $C_i$ | Count for the *i*-th diagnostic pulse |
| MZM | Mach–Zehnder modulator | | |
| PBS | Polarization beam splitter | | |
| PD | Photodiode | | |
| PID | Proportional–integral–derivative | | |
| PINPD | P-i-n photodiode | | |
| PM | Phase modulator | | |
| QBER | Quantum bit error rate | | |
| QKD | Quantum key distribution | | |
| SL | Storage line | | |
| SPD | Single-photon detector | | |
| TLD | Tunable laser driver | | |
| VOA | Variable optical attenuator | | |

## 2. Method

As shown in Figure 1a, the output intensity of the MZM is distorted by the phase (bias) drift $\theta_{drift}$, which is mainly caused by temperature changes and the photorefractive effect [30,50]. This negatively affects the performance of a QKD system that requires precisely defined pulse intensities. In this study, we propose a bias control method that compensates for it in the post-processing step after real-time bias drift estimation using

diagnostic pulses. Figure 1b shows the block diagram of the method applied to a decoy-state BB84 QKD system [7,27]. The detailed procedure is as follows. Here, the transmitter and receiver are indicated as Alice and Bob, respectively, similar to conventional QKD systems.

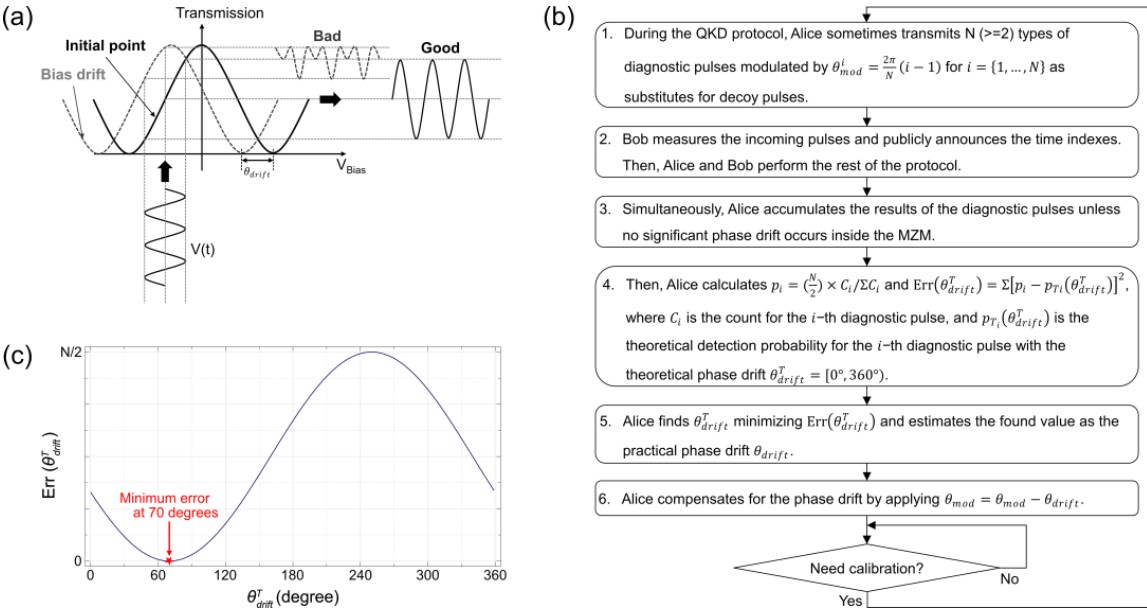

**Figure 1.** (**a**) Transfer functions of the Mach–Zehnder modulator (MZM) with and without a transmission distortion owing to the bias drift $\theta_{drift}$; (**b**) Block diagram of the proposed method applied to a decoy-state BB84 quantum key distribution (QKD) system; (**c**) Calculated $\mathrm{Err}\left(\theta_{drift}^T\right)$ according to the theoretical phase drift $\theta_{drift}^T$. The error is minimized when the $\theta_{drift}^T$ maximally matches the practical value $\theta_{drift}$. Thus, Alice can estimate $\theta_{drift} = 70°$ from the minimum error at $\theta_{drift}^T = 70°$.

1. During the QKD protocol, Alice prepares $N$ ($\geq 2$) types of diagnostic pulses whose MZM phases are modulated by $\theta_{mod}^i = \frac{2\pi}{N}(i-1)$ with uniformly distributed probabilities for $i = \{1, \ldots, N\}$. Thereafter, Alice sometimes transmits them to Bob as substitutes for the decoy pulses. Similar to the signal and decoy pulses, the diagnostic pulses are attenuated to single-photon levels. This method does not weaken the security of the QKD significantly because nobody except Alice can distinguish between the decoy and diagnostic pulses. Conventionally, the MZM output intensity can be described as [34,35,39,51,52]:

$$I_{out} = L_{in} I_{in} \cos^2\left(\frac{\theta_{mod} + \theta_{drift}}{2}\right) \qquad (1)$$

where $\theta_{mod}$ and $\theta_{drift}$ are the phase modulation and phase drift, respectively, $L_{in}$ is the insertion loss and $I_{in}$ is the input intensity.

2. Bob receives and measures the incoming pulses using single-photon detectors (SPDs). After measuring, he publicly announces the time indexes where the signals are detected. Thereafter, Alice and Bob perform the remaining protocols, such as key sifting, error correction and privacy amplification.

3. Simultaneously, Alice accumulates the detection results of the diagnostic pulses unless there are no significant phase drifts. The optimal accumulation time strongly depends on the ambient environment, channel loss, detection efficiencies and pulse intensities.

4. After accumulation, Alice calculates the normalized detection probabilities $p_i$ as [53,54]

$$p_i = \left(\frac{N}{2}\right) \times C_i / \Sigma C_i, \qquad (2)$$

where $C_i$ is the count for the $i$-th diagnostic pulse. $C_i$ can be easily obtained from Bob's announcement. Subsequently, Alice builds the following error model based on the least-squares method [53,54]:

$$\text{Err}\left(\theta_{drift}^{T}\right) = \Sigma\left[p_i - p_{Ti}\left(\theta_{drift}^{T}\right)\right]^2, \tag{3}$$

where $p_{T_i}\left(\theta_{drift}^{T}\right) = \cos^2\left(\frac{\theta_{mod}^i + \theta_{drift}^T}{2}\right)$ is the theoretical detection probability for the $i$-th diagnostic pulse with corresponding phase value $\theta_{mod}^i$ and theoretical phase drift $\theta_{drift}^{T} = \left[0°,\ 360°\right)$. Alice may calculate the theoretical values each time the probability is examined or may use a look-up table that has been created in advance.

5. Alice finds $\theta_{drift}^{T}$ minimizing $\text{Err}\left(\theta_{drift}^{T}\right)$ by adjusting $\theta_{drift}^{T}$ from 0–360°. Subsequently, the found value is estimated as $\theta_{drift}$ because the error is minimized when $\theta_{drift}^{T}$ maximally matches the practical value $\theta_{drift}$. For example, Alice can assume $\theta_{drift} = 70°$ with the smallest $\text{Err}\left(\theta_{drift}^{T}\right)$ at $\theta_{drift}^{T} = 70°$, as shown in Figure 1c. The estimation accuracy depends on the adjustment interval of $\theta_{drift}^{T}$. As the interval becomes smaller, the prediction accuracy becomes better. However, more computational power and time are required.

6. Finally, Alice compensates for the estimated phase drift by applying $\theta_{mod} = \theta_{mod} - \theta_{drift}$; therefore, the MZM bias point is maintained at the desired point. Accordingly, as the $\theta_{drift}$ term of the MZM output intensity is erased, Equation (1) becomes

$$\begin{aligned} I_{out} &= L_{in} I_{in} \cos^2\left(\tfrac{\theta_{mod} - \theta_{drift} + \theta_{drift}}{2}\right), \\ &= L_{in} I_{in} \cos^2\left(\tfrac{\theta_{mod}}{2}\right). \end{aligned} \tag{4}$$

Alice repeats the above calibration steps at specific intervals or when a phase drift higher than a predefined threshold is detected.

## 3. Experimental Results

We performed experiments in a laboratory and in the field. The experimental setup for the laboratory test is shown in Figure 2. A 1550 nm distributed feedback (DFB) laser was used for generating continuous-wave (CW) light. The MZM used in the experiment was an AZ-0S5-10-PFU-PFU (EOSpace Inc., USA) model, which utilizes a Z-cut LiNbO$_3$ crystal optimized for a wavelength of 1550 nm and has a bandwidth of up to 10 GHz. A refrigerant spray was used to abruptly reduce the temperature of the MZM. The control method was implemented on a STM32 Nucleo-144 board with a STM32F413ZH microcontroller unit (MCU). We set $N = 4$, considering the accumulation time and random bit consumption for the diagnostic signals. The desired bias point was set to the null point, and an adjustment interval of 1° was used to calculate $p_{T_i}\left(\theta_{drift}^{T}\right)$ in Equation (3). A 16-bit digital-to-analog converter and a 12-bit analog-to-digital converter built on the STM32 board were used to generate the diagnostic signals and to measure the output of the photodiode (PD).

We measured the output intensities of the MZM, which was spray-cooled using a refrigerant (SF-1013), with and without the proposed control method. As shown in Figure 2b, the output stability depended on whether the proposed method was employed or not. The output intensity with the bias control (red solid line) remained at the desired null point, whereas the intensity without the bias control (black solid line) fluctuated. Figure 2c shows the direct-current (DC) bias voltage (blue solid line) used to compensate for the phase drifts, which were estimated using the MZM output intensities of the four diagnostic signals, as shown in Figure 2d. The phase drift (Step 4 in Section 2) was estimated from the output voltage of the PD instead of the photon counts of the SPD because CW light was

used in the laboratory test. The fluctuation near 3.5 min in Figure 2c was attributed to an MZM $V_{2\pi}$ of approximately 7 V.

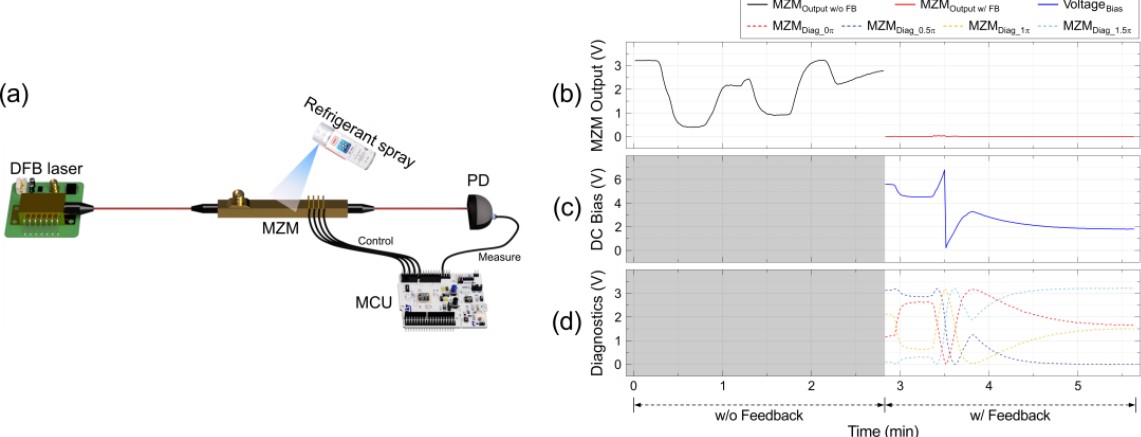

**Figure 2.** Experimental setup and results of the laboratory tests: (**a**) Experimental setup of the laboratory test; (**b**) MZM output intensity of signal modulation. The black and red solid lines represent the output intensities with and without the proposed control method, respectively; (**c**) Direct-current (DC) bias voltage to compensate for the bias drift; (**d**) MZM output intensities of the diagnostic modulations. The abbreviations are defined as follows: distributed feedback laser (DFB laser); photodiode (PD); and microcontroller unit (MCU).

In the field test, we applied the control method to a real testbed, namely, a $1 \times 3$ QKD network system installed in a secure communication system of a smart factory in South Korea. The field deployment of the QKD network system and its corresponding experimental setup are shown in Figure 3. The control system was implemented using a personal computer and an FPGA board equipped with multiple 16-bit digital-to-analog converters, instead of the STM32 board used in the laboratory test. Although different hardware platforms were used, parameters such as $N$, the step interval and the desired setpoint had similar values in both tests. Each transmitter (Alice) had an individual control system and performed the steps described in Section 2 at the respective time slots allocated via time-division multiplexing. We set the count accumulation time of the diagnostic pulses to approximately 5 min because there were no significant phase drifts during a 5 min window (in the facility's communication room); this means that the bias drift was compensated every 5 min. As mentioned, the optimal accumulation time depends on the QKD system parameters, such as temperature, channel loss, detection efficiency and pulse intensity.

The field test results are shown in Figure 4. We measured the extinction ratios (ERs) of the MZMs and QKD performance parameters, such as sifted key rates and quantum bit error rates (QBERs), over 4–5 days. Figure 4a,b show the sifted key rates and QBERs of the $1 \times 3$ QKD network, respectively. Figure 4c shows the ERs of the MZMs. During the test period, all the ERs were maintained over 20 dB, and satisfactory QKD performances were achieved. There was no significant difference in the stabilities despite the different counts owing to the different channel losses. These results indicate that the proposed method could maintain MZM stability in an actual network environment.

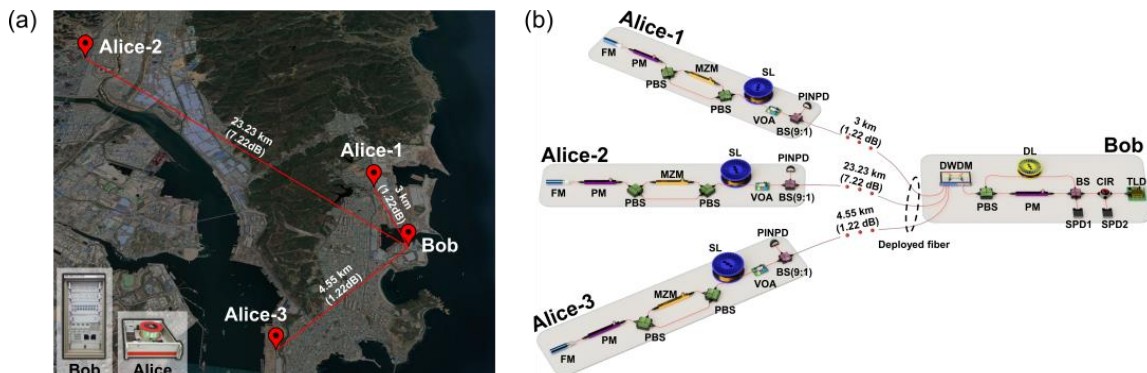

**Figure 3.** Field deployment and experimental setup of the 1 × 3 quantum key distribution (QKD) network system: (**a**) Field deployment in a smart factory in South Korea. Map data: Google, © 2022 Maxar Technologies, TerraMetrics. The insets show the equipment for the transmitter (Alice) and receiver (Bob); (**b**) Experimental setup. Time- and wavelength-division multiplexing were used to establish the 1 × 3 network. The lengths (in km) and losses (in dB) of the quantum channels were indicated. The abbreviations are defined as follows: tunable laser driver (TLD); circulator (CIR); beam splitter (BS); single-photon detector (SPD); phase modulator (PM); delay line (DL); polarization beam splitter (PBS); dense wavelength division multiplexer (DWDM); p-i-n photodiode (PINPD); variable optical attenuator (VOA); storage line (SL); and Faraday rotator mirror (FM).

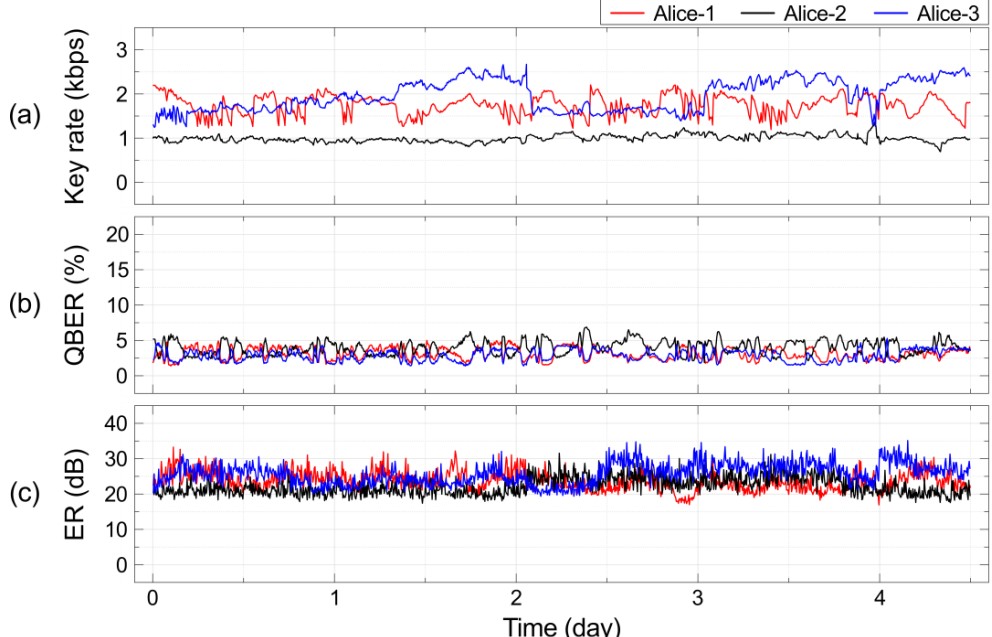

**Figure 4.** Experimental results of the field test: (**a**) Sifted key rates; (**b**) Quantum bit error rates (QBERs); (**c**) Extinction ratios (ERs) of the MZMs. The red, black and blue solid lines are the results of Alice 1–3, respectively.

## 4. Conclusions

We proposed and experimentally demonstrated an efficient MZM bias control method for QKD systems. The proposed method is cost-effective and simplifies the transmitter. This is because it can be implemented using a software modification for bias drift estimation without any additional hardware. This is a crucial advantage of the proposed method, particularly in a QKD network system comprising many users. Unlike the PID control method that converges to the desired setpoint by gradually minimizing errors, the proposed method immediately compensates for the phase drift estimated through the diagnostic pulses. Thus, based on our own experience, it can attain the setpoint faster than the

conventional PID control method. In addition, the proposed method can be implemented on a parallel processor, such as an FPGA. Finally, there is no significant decrease in the key rate, as only the time slots allocated to the existing decoy pulses are used for diagnostic pulses.

We demonstrated the implementation feasibility of the proposed method in a laboratory test and in a field test in a $1 \times 3$ QKD network testbed, which was installed in the security facility of a smart factory in South Korea. In the experimental results, we showed that the proposed method could handle temperature changes and maintained the ERs over 20 dB (bit error rate $\leq 1\%$) for several days, even in an actual network environment. Although the measured ERs did not meet the maximum performance that was provided by the manufacturer (i.e., 30 dB), they were sufficient for application in the QKD system (considering that a maximum error of only 1% was attributed to the 20 dB ER). Furthermore, they can be further improved by optimizing the compensation period, diagnostic pulse proportion and $N$. As mentioned, the optimal control period strongly depends on the implementation environment. Additionally, it has a trade-off relation with the key generation rate. Thus, to optimize the QKD performance, future researchers should take this into account. As we perceived that the real-time stabilities of the ERs, sifted key rates and QBERs sufficiently showed the feasibility of the proposed method, we did not calculate the secret key rate indicating the overall QKD performance in this study. However, we assume that it is not significantly different from that of the conventional decoy-state BB84 QKDs. This is because our method can be implemented by consuming only the time slots of existing decoy pulses.

Based on our experimental results, we believe the proposed method can provide an efficient way to implement an MZM bias control in QKD systems. In the future, we will improve control performance via parameter optimization. In addition, although the current method in this study corresponds to only single MZM bias control, we will develop an advanced one for multiple MZMs for a wide range of applications, such as coherent optical communications. A rigorous security analysis will also be conducted.

**Author Contributions:** Conceptualization, C.-H.P. and M.-K.W.; methodology, C.-H.P.; software, C.-H.P. and B.-K.P.; validation, S.-W.J., H.J., S.K. and S.-W.H.; formal analysis, C.-H.P.; investigation, S.-W.J. and H.J.; resources, S.K. and S.-W.H.; data curation, C.-H.P.; writing—original draft preparation, C.-H.P.; writing—review and editing, S.-W.H.; visualization, S.-W.H.; supervision, S.-W.H.; project administration, S.-W.H.; funding acquisition, S.-W.H. All authors have read and agreed to the published version of the manuscript.

**Funding:** This research was funded by the National Research Foundation of Korea, grant numbers 2019M3E4A1079777 and 2021M1A2A2043892; by the Institute for Information and Communications Technology Promotion, grant numbers 2020-0-00972 and 2020-0-00947; and by the Korea Institute of Science and Technology, grant number 2E31531.

**Data Availability Statement:** The data presented in this study are available on request from the corresponding author.

**Conflicts of Interest:** The authors declare no conflict of interest.

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
