# Peer review of "Experimental Demonstration of an Efficient Mach–Zehnder Modulator Bias Control for Quantum Key Distribution Systems"

_electronics, doi:10.3390/electronics11142207_

Round 1

Reviewer 1 Report

Implementation of an efficient Mach–Zehnder modulator bias control for quantum key distribution systems

Authors: Chang Hoon Park, et al

This manuscript proposed and experimentally demonstrated an efficient MZM bias control method that can be effectively  applied to QKD systems.

The authors’ claims that the proposed scheme is robust against imperfections during the transmission via a software modification for bias drift estimation without additional hardware.  They have shown that the improvement in the scaling of the QKD network system by comprising many users.

 This manuscript is timely and contributes significant information to the ongoing study of MZM in the practical integration of QKD in high connectivity networks.  The organization of the paper is sensible and it allows the reader to get familiar with the concept of MZM bias control method for QKD systems. My main remark on the paper is the fact that current devices rely on bulky and high power demanding phase modulation which hinder the sought-scaling and energy efficiency. The authors must clearly discuss how much they achieve the secure key rates and state the proposal will also be beneficial to advanced coherent optical communications.  

The authors ignore some important reference in the introduction: on the recent works on how phase modulation of coherent states play in quantum communication channels like in the paper https://journals.aps.org/pra/abstract/10.1103/PhysRevA.92.012317 .

and the new encoding/decoding method when the emitted states are coherent states. The method of using non-determinstic noiseless linear amplifiers to improve the protocol performance against the phase diffusion effect in quantum channels like in the papers https://doi.org/10.1364/JOSAB.36.002938 where the information is coded on phase shifts and at the decoding stage https://journals.aps.org/pra/abstract/10.1103/PhysRevA.93.062315 and the authors are expected to raise the level of the manuscript by referring to them appropriately.

As the minor suggestions, authors may also address the demands on the single-photon source, which equivalently used by Alice and Bob. Can practical single-photon sources be more promising for efficient MZM bias control method?

I think this paper is a worthy contribution for Electronics, and I recommend it for publication after the above remarks are addressed. 

Reviewer 2 Report

1. The idea of the research topic is excellent but poor representation

2. A section named "Literature Review" should be added after "Introduction".  In the "Literature Review" section add as much as current research works you can relate to your work. Try including research ideas of other papers related to you along with a comparison with your research work.

3. No clear Methodology was found and mentioned at the time of reviewing your article. You should add a separate section named "Methodology" and should mention your entire method and purpose here as the purpose of your paper is not also clear by your description.

4. The conclusion is not clear and please add some future works in the conclusion or make a separate section named "Future Work" and add some future commitment to implement. 

5. Paper should be submitted without author information and affiliation; after CRC you can submit the author information and affiliation

Reviewer 3 Report

please see my reviewer report

Round 2

Reviewer 2 Report

-Comments and review indicated previously, doesn't incorporate properly. 

Reviewer 3 Report

Authors replied effectively to all comments. Accept